# Mitochondrial Targeting of Antioxidants Alters Pancreatic Acinar Cell Bioenergetics and Determines Cell Fate

**DOI:** 10.3390/ijms20071700

**Published:** 2019-04-05

**Authors:** Jane A. Armstrong, Nicole J. Cash, Jack C. Morton, Alexei V. Tepikin, Robert Sutton, David N. Criddle

**Affiliations:** 1Department of Molecular and Clinical Cancer Medicine, Institute of Translational Medicine, University of Liverpool, Liverpool L69 3BX, UK; janearm@liverpool.ac.uk (J.A.A.); sutton@liverpool.ac.uk (R.S.); 2Department of Cellular & Molecular Physiology, Institute of Translational Medicine, University of Liverpool, Liverpool L69 3BX, UK; Nicole.Cash@liverpool.ac.uk (N.J.C.); jmorton@liverpool.ac.uk (J.C.M.); kiev@liverpool.ac.uk (A.V.T.)

**Keywords:** oxidative stress, mitochondrial dysfunction, MitoQ, DecylTPP, Seahorse, pancreatic acinar cell, antioxidants, mitochondrial targeting

## Abstract

Mitochondrial dysfunction is a core feature of acute pancreatitis, a severe disease in which oxidative stress is elevated. Mitochondrial targeting of antioxidants is a potential therapeutic strategy for this and other diseases, although thus far mixed results have been reported. We investigated the effects of mitochondrial targeting with the antioxidant MitoQ on pancreatic acinar cell bioenergetics, adenosine triphosphate (ATP) production and cell fate, in comparison with the non-antioxidant control decyltriphenylphosphonium bromide (DecylTPP) and general antioxidant *N*-acetylcysteine (NAC). MitoQ (µM range) and NAC (mM range) caused sustained elevations of basal respiration and the inhibition of spare respiratory capacity, which was attributable to an antioxidant action since these effects were minimal with DecylTPP. Although MitoQ but not DecylTPP decreased cellular NADH levels, mitochondrial ATP turnover capacity and cellular ATP concentrations were markedly reduced by both MitoQ and DecylTPP, indicating a non-specific effect of mitochondrial targeting. All three compounds were associated with a compensatory elevation of glycolysis and concentration-dependent increases in acinar cell apoptosis and necrosis. These data suggest that reactive oxygen species (ROS) contribute a significant negative feedback control of basal cellular metabolism. Mitochondrial targeting using positively charged molecules that insert into the inner mitochondrial member appears to be deleterious in pancreatic acinar cells, as does an antioxidant strategy for the treatment of acute pancreatitis.

## 1. Introduction

Mitochondrial dysfunction is a central event in the development of acute pancreatitis, a severe and painful disease which carries a significant mortality and lacks a specific therapy [1,2,3]. Acute pancreatitis is a major cause of gastrointestinal hospital admissions and imposes a significant socio-economic burden [4]. A role for reactive oxygen species (ROS) has been identified in the development of acute pancreatitis on the basis of an elevated oxidative status and reduced antioxidant capacity observed in both clinical studies and experimental animal models [5,6]. Despite early promise, clinical trials of antioxidants in acute pancreatitis have proved largely unsuccessful [7,8]. The use of antioxidant therapy in general is controversial and has proven ineffective in various pathologies [5,9] with evidence that antioxidants can exert adverse effects per se, including the promotion of melanoma metastasis [10]. In isolated pancreatic acinar cells, the inhibition of bile acid-induced ROS generation by the general antioxidant *N*-acetylcysteine (NAC) promoted necrosis at the expense of apoptotic cell death [6,11,12] supporting an important role for ROS in defining acinar cell fate. More recently, we have demonstrated that the direct application of oxidants altered pancreatic acinar cell bioenergetics, resulting in a cyclophilin D-independent shift from apoptotic to necrotic cell death [13].

The targeting of antioxidants to mitochondria is a recent approach for the amelioration of diseases in which mitochondrial dysfunction is integral [14]. Of particular interest is the compound MitoQ, which has been shown to exert beneficial actions in multiple pathologies including ischaemia-reperfusion injury [15,16], colitis [17], encephalomyelitis [18], diabetes [19] and sepsis [20] and in clinical trials of Parkinson’s, liver and vascular diseases [21,22,23]. It comprises a triphenylphosphonium cation linked to coenzyme Q_10_ and accumulates in the hydrophobic core of the phospholipid bilayer on the matrix-facing surface of the inner mitochondrial membrane [24,25]. Uptake into the organelle is driven by the mitochondrial membrane potential [26] and MitoQ is thought to be continually recycled to active antioxidant ubiquinol by complex II of the respiratory chain [27]. High localised antioxidant concentrations are achieved; however, there may also be consequences of the introduction of a charged acyl moiety into the mitochondrial membrane that are independent of antioxidant action [28,29]. Previously, we have shown that MitoQ was largely non-protective in in vivo experimental AP models, demonstrating that a mixed profile [30] and a deeper understanding of targeting antioxidants to mitochondria in pancreatic acinar cells is important.

The aim of this study was to investigate the effects of antioxidant mitochondrial targeting on pancreatic acinar cell bioenergetics and cell fate. A comparison of MitoQ, its non-antioxidant control DecylTPP (the structures of which have previously been described [28]) and general antioxidant NAC was performed using Seahorse XF24 analysis of respiratory function, confocal microscopy of NADH/FAD^+^ levels and plate-reader assays of cellular ATP level and cell death modalities. Our results demonstrate that the targeting of antioxidants to mitochondria induced specific and non-specific changes to bioenergetics that had important consequences for pancreatic acinar cell health.

## 2. Results

### 2.1. Differential Effects of MitoQ and DecylTPP on Acinar Cell Bioenergetics

Application of MitoQ and its non-antioxidant control DecylTPP (1–10 µM) caused distinct alterations of oxygen consumption rate (OCR) in isolated pancreatic acinar cells (Figure 1A–C). Use of the respiratory function (“stress”) test, involving the sequential addition of inhibitors of the electron transport chain, revealed important differences between MitoQ and DecylTPP on mitochondrial bioenergetics. No significant changes of respiratory parameters occurred with either MitoQ or DecylTPP at 1 µM. Higher levels of MitoQ (5–10 µM) but not DecylTPP increased basal respiration in a concentration-dependent manner, an effect commencing within 5 min of application and continuing over a 30 min period (Figure 1B,C and Figure 2A,B). In contrast, both MitoQ and DecylTPP (5–10 µM) caused significant elevations of the extracellular acidification rate (ECAR) within 5 min which were sustained over 30 min. At the highest concentration (10 µM), MitoQ effects were significantly greater than those caused by 10 µM DecylTPP at both 5 and 30 min (Figure 2C,D). Additional experiments using confocal microscopy showed that neither MitoQ nor DecylTPP caused a significant elevation of basal ROS at 10 µM (Appendix A).

The mitochondrial ATP turnover capacity, visualised by the addition of oligomycin (Figure 1A,C), was greatly decreased by both MitoQ and DecylTPP at 5 and 10 µM (Figure 2E), suggesting a non-specific action of mitochondrial targeting. This inhibition of ATP turnover capacity was coupled with a marked elevation of proton leak with both MitoQ and DecylTPP (Figure 2F). Furthermore, MitoQ greatly reduced the spare respiratory capacity at 10 µM, in contrast to DecylTPP which showed no effect (Figure 2G).

### 2.2. Inhibitory Effects of MitoQ on the NADH/FAD^+^ Redox Ratio

Confocal microscopy experiments examined the effects of MitoQ and DecylTPP on mitochondrial bioenergetics by measuring NADH and FAD^+^ autofluorescence levels dynamically in single pancreatic acinar cells; a typical mitochondrial distribution of NADH and FAD^+^ was observed, as reported previously [13] (Figure 3A). Addition of 1 μM MitoQ induced a significant time-dependent NADH decrease that was mirrored by a FAD^+^ increase, in contrast to DecylTPP which caused no significant changes above controls (Figure 3B,C). The protonophore carbonyl cyanide m-chlorophenyl hydrazone (CCCP), added after a 30-min application of MitoQ/DecylTPP, was used to establish a maximal effect on the NADH/FAD^+^ redox ratio (Figure 3C).

### 2.3. Effects of N-Acetylcysteine on Acinar Cell Bioenergetics

Further experiments were performed to assess the effects of the general, non-mitochondrial-targeted antioxidant NAC on bioenergetics in isolated pancreatic acinar cells (Figure 4A–C). In contrast to MitoQ, which exerted effects at µM concentrations, much greater levels (mM) of NAC were required to alter cellular bioenergetics. At 1 mM, NAC did not significantly alter basal respiration; however, at concentrations ≥5 mM, it significantly increased baseline OCR, similar to the elevations observed with MitoQ. This effect gradually increased over time and persisted at 30 min after application (Figure 4D,E), accompanied by a significant increase in ECAR (Figure 4F). At 10 mM, NAC induced a modest but statistically significant reduction of ATP turnover capacity (Figure 5G) and a marked increase in proton leak at both 5 and 10 µM (Figure 4H). In contrast to MitoQ, which produced a decrease of spare respiratory capacity (SRC) only at the highest concentration evaluated (10 µM), NAC induced a dramatic concentration-dependent reduction of SRC, which commenced at 1 mM and was maximal at 10 mM (Figure 4H).

### 2.4. Effects of MitoQ and DecylTPP on Cell Death, ATP Production and Cell Viability

The application of MitoQ and DecylTPP (1–10 µM) to pancreatic acinar cells elicited a cell death pathway activation that varied according to the severity of insult. Rapid elevations of both apoptosis and necrosis were detected in response to both MitoQ and DecylTPP at 10 µM (Figure 5A,B). However, at lower levels (1 µM), only the mitochondrial-targeted antioxidant MitoQ induced necrosis at all time-points (Figure 5A). In contrast, elevated necrosis was evident only at 13 h after the application of 1 µM DecylTPP. Similar differences were detected in apoptosis measurements; 1 µM MitoQ significantly increased apoptosis at 2.5 h, whereas 1 µM DecylTPP produced a significant change only at 13 h (Figure 5B). Similarly, in separate plate reader assays, MitoQ decreased cell viability at all concentrations (1, 5 and 10 µM) in a concentration-dependent manner, whereas DecylTPP caused a significant change only at the highest concentration (Figure 5C).

In order to determine whether the detrimental changes of bioenergetics induced by antioxidants in acinar cells were linked to a reduction of cellular ATP levels, separate luciferase-based plate-reader assays were conducted. Applications of MitoQ or DecylTPP (1–10 µM) to pancreatic acinar cells both caused concentration-dependent decreases of cellular ATP at ≥5 µM (Figure 4D).

### 2.5. Effects of NAC on Cell Death, ATP Production and Cell Viability

Application of NAC (1–10 mM) elicited concentration-dependent pancreatic acinar cell death, with necrosis induced at earlier time-points and at lower concentrations preferentially over apoptosis. Thus, a rapid elevation of necrosis was induced by 10 mM NAC within 2.5 h (Figure 6A), whereas apoptosis was not different from control. At 7 h, both 5 and 10 mM NAC induced necrosis, whereas only the higher concentration caused a significant rise of apoptosis. Lower levels of NAC ≤5 mM induced significant alterations of necrosis only at the later time-point of 13 h (Figure 6A), without a concomitant induction of apoptosis (Figure 6B).

In separate experiments, the application of NAC induced a concentration-dependent decrease of pancreatic acinar cell viability measured with lactate dehydrogenase (LDH), although this was significant only at the highest concentration (10 mM; Figure 6C). In luciferase-based plate-reader assays, 10 mM NAC caused a decrease of cellular ATP in pancreatic acinar cells (Figure 6D), in agreement with a reduction of ATP turnover capacity observed in OCR measurements at this concentration (Figure 4G).

## 3. Discussion

The results of this study show that the mitochondrial-targeted antioxidant MitoQ induced significant changes of pancreatic acinar cell bioenergetics that resulted in diminished ATP production and cell death. Some actions were shared by the general antioxidant NAC, albeit at higher concentrations (mM compared to µM), indicating specific effects on respiration due to the scavenging of ROS that may be accentuated by mitochondrial targeting. Thus, a marked elevation of basal respiration in pancreatic acinar cells occurred rapidly in response to MitoQ and was maintained over a 30-min application; this effect was shared by NAC, although it was less pronounced than with the targeted compound. Our results are therefore consistent with a recent study in proximal tubule-derived opossum kidney cells that demonstrated an increase in basal respiration in response to MitoQ [31].

Although ROS are capable of inflicting cellular damage, they are a natural consequence of physiological electron transport chain activity in respiring cells. Superoxide (O_2_^−^) is generated at complexes I and III, then converted to peroxynitrite, hydrogen peroxide (H_2_O_2_) and the hydroxyl radical (OH^−^) [32,33]. Despite a predominant focus on their detrimental effects, recent evidence demonstrates that ROS participate in physiological signalling events [34,35,36]. A fine balance is usually maintained between the production and quenching of cellular ROS, with excesses prevented by an endogenous homeostatic antioxidant system. Previously, we have shown that the application of exogenous H_2_O_2_ to pancreatic acinar cells caused a decrease of basal respiration [13]. Quenching of mitochondrial ROS by MitoQ or NAC was found to counteract a natural, endogenous depression of basal respiration in acinar cells. ROS may provide negative feedback control of basal respiration as this effect was far less evident with DecylTPP. Superoxide has been shown to directly activate uncoupling proteins [37] that would inhibit basal respiration; in particular, UCP-2 is thought to exert a homeostatic role in the regulation of mitochondria-derived ROS [38,39] that may involve the alteration of mitochondrial glutathione levels [40] and is overexpressed in models of acute pancreatitis [41]. Alternatively, the increased respiration may also relate to the ability of MitoQ to affect mitochondrial Ca^2+^ homeostasis, but again it is likely linked to an action of ROS since NAC had a similar effect on basal respiration. A study in HeLa cells showed that µM concentrations of MitoQ increased mitochondrial Ca^2+^ by inhibiting efflux from the organelle [42]. Brief elevations of mitochondrial Ca^2+^ in pancreatic acinar cells would boost respiration via activation of Ca^2+^-dependent dehydrogenases leading to increased TCA cycle activity [43,44]. 

Hyperstimulation could also impose significant stress on mitochondria, leading to organelle dysfunction. In accordance with this, excessive antioxidant action in skin and lung cells led to reductive stress which induced more severe damage than that inflicted by H_2_O_2_ [45]. Here, in pancreatic acinar cells, MitoQ caused a reduction of the spare respiratory capacity—an action shared by NAC but not by DecylTPP—suggesting a specific detrimental antioxidant action on maximal respiration. This is consistent with mitochondrial depolarisation elicited by MitoQ in pancreatic acinar cells at the same concentrations [46], and a reduction of ∆Ψ_m_ in breast and lung and cancer cell lines [29]. Spare respiratory capacity is considered a vital reserve of mitochondria in the response to stress but is reduced in pathophysiological conditions including cardiac and neurodegenerative damage [47,48]; antioxidants, by reducing this capacity in pancreatic acinar cells, may limit their capability to respond to adverse conditions. MitoQ caused a sustained reduction of cellular NADH levels, similar to toxic changes seen in response to precipitants of acute pancreatitis, including cholecystokinin hyperstimulation, bile acid and non-oxidative ethanol metabolites, that ultimately lead to the loss of ATP production and pancreatic acinar cell necrosis [11,43,49]. The inhibition of acinar cell bioenergetics found in the present study is consistent with our previous work in which MitoQ did not protect against bile acid-induced acute pancreatitis and showed mixed results in caerulein-induced acute pancreatitis [46].

Our present study has also identified non-specific effects of the mitochondrial targeting of MitoQ. Thus, a dramatic reduction of the ATP turnover capacity was elicited by MitoQ; this change, detected via the oxygen consumption rate, was reflected in diminished cellular ATP levels and associated with development of pancreatic acinar cell necrosis. Since the inhibition of ATP production was shared by DecylTPP, this would indicate a non-specific action of targeting of the charged acyl moiety to mitochondria. Our results are consistent with a near complete inhibition of ATP turnover capacity elicited by both MitoQ and DecylTPP shown in a kidney MES-13 cell line [28]. It was proposed that the DecylTPP moiety undergoes rapid cycling between the mitochondrial matrix and cytosol, thereby uncoupling mitochondria, consistent with inhibition of ATP turnover capacity. The effect we observed in pancreatic acinar cells was accompanied by a marked elevation of proton leak with both MitoQ and DecylTPP, suggesting a compromise of mitochondrial membrane integrity by insertion of the DecylTPP moiety. DecylTPP has recently been shown to directly increase the permeability of artificial liposomes; in that study, both MitoQ and DecylTPP caused mitochondrial membrane depolarisation and a swelling of kidney tissue that was dependent on the acyl chain [31]. 

Interestingly, a robust increase of pancreatic acinar cell ECAR in response to MitoQ, DecylTPP and NAC occurred concomitantly with the inhibition of ATP turnover, suggesting a compensatory shift in metabolism towards glycolysis when oxidative phosphorylation was inhibited. An impairment of oxidative phosphorylation coincident with an increased ECAR in response to MitoQ has recently been shown in MDA-MB-231 cells suggesting a switch in energy production to glycolysis in this cell line [29]. However, despite a potential boost of ATP production by this route, a decrease of total cellular ATP occurred in response to all three agents that was associated with induction of pancreatic acinar cell death, notably necrosis. All three agents induced a fall in cellular ATP levels; NAC caused a preferential increase in necrosis over apoptosis, with significant effects appearing at lower concentrations and at earlier time-points in the former pathway; both MitoQ and DecylTPP also caused increases in apoptosis and necrosis that were dependent on concentration. It should be noted that the concentrations of MitoQ investigated in the present study may be higher than those achieved during therapeutic application. Previous studies have suggested that the intra-mitochondrial concentration of MitoQ was approximately 0.7 µM in mice fed 500 µM MitoQ in drinking water for 4–6 months [27], while an intravenous injection of 750 nM MitoQ was well-tolerated, although toxic effects were evident at 1 µM [50].

The present findings with antioxidants are consistent with prior evidence that ROS preferentially promote apoptosis in pancreatic acinar cells [6] and that bile acid-induced, Ca^2+^-dependent necrosis was potentiated by NAC with a concomitant reduction of apoptosis [11]. The balance between apoptosis and necrosis in the exocrine pancreas may play a crucial role in the development of acute pancreatitis [11,51,52,53]. Taken together, the actions of MitoQ on cellular bioenergetics observed in isolated pancreatic acinar cells are in accord with the lack of protective effects observed in in vivo acute pancreatitis models [46] and indicate the need for caution when considering mitochondrial antioxidant targeting for the treatment of acute pancreatitis. 

## 4. Materials and Methods

### 4.1. Animals and Preparation of Isolated Pancreatic Acinar Cells

Fresh pancreatic acinar cells were isolated using standard collagenase (Worthington Biochemical Corporation, Lakewood, NJ, USA) from the pancreata of young (8–12 weeks old) adult C57BL/6 (wild type) as previously described [1,11]. The animals were humanely sacrificed by cervical dislocation (Schedule 1 procedure) in accordance with the Animals (Scientific Procedures) Act (1986) under Establishment Licence 40/2408 and with approval by the University of Liverpool Animal Welfare Committee and Ethical Review Body (AWERB; 10/12/2013). The extracellular solution contained (mM): 140 NaCl, 4.7 KCl, 1.13 MgCl_2_, 1 CaCl_2_, 10 d-glucose, and 10 HEPES (adjusted to pH 7.25 using NaOH).

### 4.2. Confocal Microscopy

Confocal imaging was performed using a Zeiss LSM510 system (Carl Zeiss Jena GmbH, Jena, Germany). Mitochondrial metabolism was assessed in unloaded cells by NADH (excitation was 363 nm and emission at 390–450 nm) and FAD^+^ (excitation was 458 nm and emission at 505–560 nm) autofluorescence simultaneously. The redox ratio was determined by calculating the ratio of the measured fluorescence intensities of NADH and FAD^+^ [54,55]. Fluorescence measurements were expressed as changes from basal fluorescence (*F*/*F*_0_ ratio), where *F*_0_ represents the initial fluorescence recorded at the start of the experiment and *F* the fluorescence recorded at specific time points (“*n*” represents the number of cells studied for each experimental protocol).

### 4.3. Detection of Apoptotic and Necrotic Cell Death Pathways

For the detection of necrosis and apoptosis, a POLARstar Omega fluorescence microplate reader (BMG Labtech, Ortenberg, Germany) was employed for time-course experiments at 37 °C. Flat-bottomed 96-well microplates (Greiner Bio-One Ltd., Stonehouse, UK) were used to seed cells at a density of 300,000 per well. Propidium iodide (PI) was used to detect necrosis and loaded at a final concentration of 10 µg/mL. Excitation was set at 520 nm and emission collection at > 590 nm. For apoptosis measurements, CellEvent^®^ Caspase-3/7 Green Ready Probes^®^ Reagent was added to the acinar cell suspension at 40 µL/mL. The excitation was 485 nm and emission at 530 nm. The fluorescence intensity was normalised to negative controls for each mouse.

### 4.4. Seahorse XF Analysis

The Seahorse XF24 Analyzer (Agilent, North Billerica, MA, USA) was used to evaluate bioenergetics function in isolated pancreatic acinar cells as reported previously [13]. OCR (oxygen consumption rate) and ECAR (extracellular acidification rate) were measured in unloaded cells, monitored in real time. Prior to bioenergetics measurements, the isolation medium was changed to unbuffered Dulbecco’s modified Eagle’s medium (DMEM, pH 7.4) supplemented with 10 mM glucose, 2 mM l-glutamine and 2 mM sodium pyruvate (Sigma-Aldrich, Gilligham, UK). The optimum number of cells/well for detection of changes in OCR and ECAR was determined to be 75,000/0.32 cm^2^. The basal OCR in pancreatic acinar cells was first measured in the absence of any additional stimulation to obtain steady-state, resting levels of respiration, predominantly driven by H^+^ flux through the F_1_F_0_-ATP synthase. A mitochondrial respiratory function “stress” test protocol was subsequently implemented to measure indices of mitochondrial function with and without antioxidants applied; oligomycin, trifluoromethoxy carbonylcyanide phenylhydrazone (FCCP), antimycin A and rotenone were injected sequentially through ports of the Seahorse Flux Pak cartridges to achieve final concentrations of 1 µg/mL, 0.3 µM and 2 µg/mL, respectively. Thus, the addition of the ATP synthase inhibitor oligomycin allowed the determination of the oxygen consumption linked to ATP production, seen as the resultant reduction of OCR. The remaining mitochondrial OCR in the presence of oligomycin is attributable to proton leak rate across the mitochondrial membrane [56]. Subsequent application of the protonophore FCCP, which increases H^+^ leak across the inner mitochondrial membrane causing a H^+^ short circuit, enabled the measurement of maximal respiration, seen as an increase in OCR. Further addition of a combination of antimycin A (Complex III inhibitor) and rotenone (Complex I inhibitor) completely blocked mitochondrial O_2_ consumption; the remaining OCR is attributable to non-mitochondrial respiration. The spare respiratory capacity, calculated as the difference between basal OCR and maximal OCR achieved after FCCP addition, indicates the ability of electron transport to respond to an increase in energy demand. This important parameter may indicate how close a cell is functioning to its bioenergetic limit and its capacity to respond to elevated stress [47,56]. The concurrent measurement of ECAR allowed an indirect determination of glycolysis; glycolytic lactate production, coupled with H^+^ release, is the predominant basis for ECAR, although recent evidence suggests that respiratory CO_2_ production may contribute [57].

### 4.5. Intracellular ATP Determination

Pancreatic acinar cells (1 × 10^6^/condition) were pretreated with antioxidant for 30 min and washed with buffer A (25 mM Tris-HCl, 10 mM KH_2_PO_4_, 150 mM KCl, 5 mM MgCl_2_, 0.1% BSA, pH 7.8). The cells were then covered and permeabilized with 200 µL 1× ATP-releasing reagent in buffer A (Sigma-Aldrich) per well for 2 min. Twenty microlitres of each supernatant was then transferred to a white plate and the measurement protocol started immediately using a POLARstar Omega Plate Reader (BMG Labtech, Germany). Eighty microlitres of mastermix, consisting of 0.3 mM luciferin potassium salt and luciferase (Sigma-Aldrich), was injected per well and the luminescence emission recorded for 15 min. Addition of the ATP synthase inhibitor oligomycin was used to show maximal blockade of mitochondrial respiration. The chemiluminescence intensity was normalised to negative controls for each mouse/run.

### 4.6. Lactate Dehydrogenase Assay

Pancreatic acinar cells were pretreated with antioxidant for 30 min, centrifuged and washed before using a Lactate Dehydrogenase Activity Assay Kit (Sigma-Aldrich). Samples were added per well and absorbance measurement started immediately using a POLARstar Omega Plate Reader (BMG Labtech, Germany). Addition of Triton X (0.5%) was used to show maximal inhibition of cell viability.

### 4.7. Reagents

The following reagents were acquired from Sigma Aldrich (reagent, Cat-no.): *N*-acetylcysteine, A7250; propidium iodide (PI), P4170; NaCl, S7653; KCl, P9333; MgCl_2_, M1028; CaCl_2_, 21114; D-glucose, 158968; HEPES, H3375; NaOH, S5881; l-glutamine, G7513; sodium pyruvate, 25030; Oligomycin, 75351; FCCP, C2920; antimycin A, A8674; rotenone, R8875; DMEM, D5030; Tris-HCl, T3253; KH_2_PO_4_, P5655; BSA, A3803; ATP-releasing reagent, FLSAR; ATP determination assay, A22066. The Lactate Dehydrogenase Activity Assay Kit, 88953 and CellEvent^®^ Caspase-3/7 Green Ready Probes^®^ Reagent, C10423 were both obtained from Thermo/Fisher. Collagenase CLSPA, LS005275 (Worthington Biochemical Corporation). MitoQ was a kind gift from Prof. M. Murphy (University of Cambridge, UK).

### 4.8. Statistical Analysis

All data were analyzed using analysis of variance (ANOVA) and Tukey’s post-hoc test with Prism 5.0 software (GraphPad Software Inc., La Jolla, CA, USA). Data are presented as mean ± S.E.M. All experiments were repeated at least three times.

## 5. Conclusions

MitoQ caused alterations of pancreatic acinar cell bioenergetics that were partly mediated by mitochondrial targeting in addition to specific antioxidant effects. Such actions are consistent with a lack of protection afforded by MitoQ in previous in vivo studies of acute pancreatitis [46], and may have implications for therapeutic use in a wider context.

## Figures and Tables

**Figure 1 ijms-20-01700-f001:**
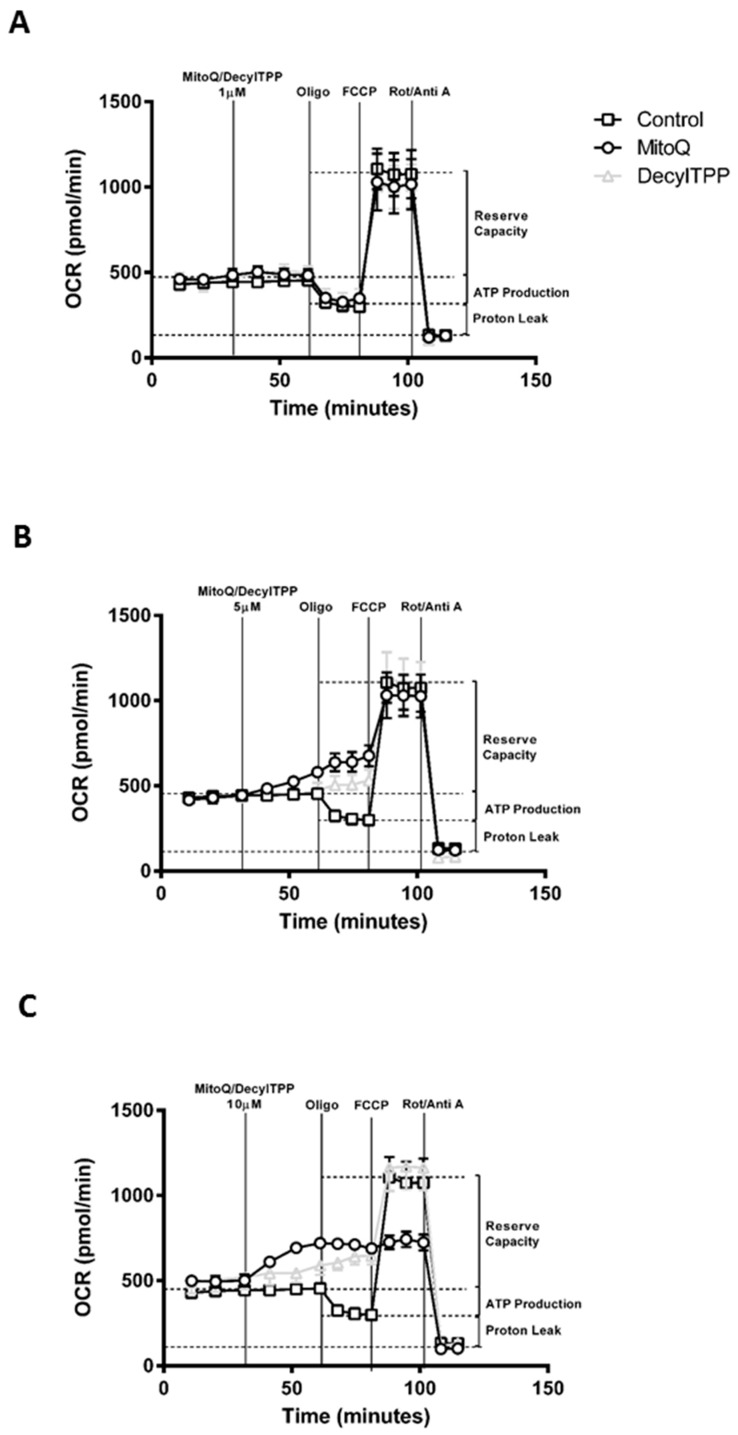
Effects of MitoQ and DecylTPP on mitochondrial bioenergetics. Plots of respiratory control experiments showing the time-dependent effects of MitoQ (open circles) and DecylTPP (open grey triangles) at (**A**) 1 µM, (**B**) 5 µM, and (**C**) 10 µM on oxygen consumption rate (OCR) changes at compared to control (open squares). A respiratory function ‘stress’ test was carried out using the sequential addition of oligomycin (Oligo, 1 μg/mL), FCCP (0.3 μM), and rotenone/antimycinA combined (Rot/AntiA, 1 μM) to detect respiratory parameters (ATP turnover capacity, proton leak and spare respiratory capacity).

**Figure 2 ijms-20-01700-f002:**
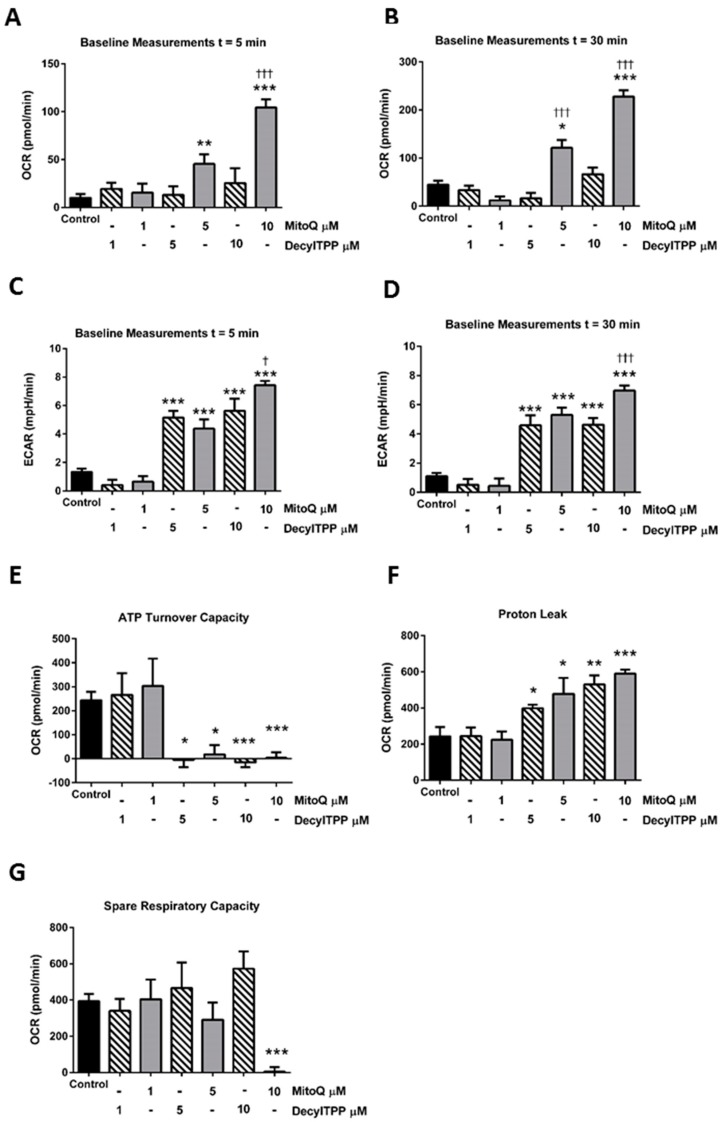
Effects of MitoQ and DecylTPP on mitochondrial bioenergetics. Bar charts showing the effects of MitoQ (black) and DecylTPP (grey) (1–10 µM) on baseline OCR changes at (**A**) 5 min and (**B**) 30 min and baseline extracellular acidification rate (ECAR) changes at (**C**) 5 min and (**D**) 30 min and (**E**) ATP turnover capacity, (**F**) proton leak and (**G**) spare respiratory capacity. Data shown as means ± SEM (*n* = 3). * *p* < 0.05, ** *p* < 0.01, *** *p* < 0.001. ^†^
*p* < 0.05, ^†††^
*p* < 0.001.

**Figure 3 ijms-20-01700-f003:**
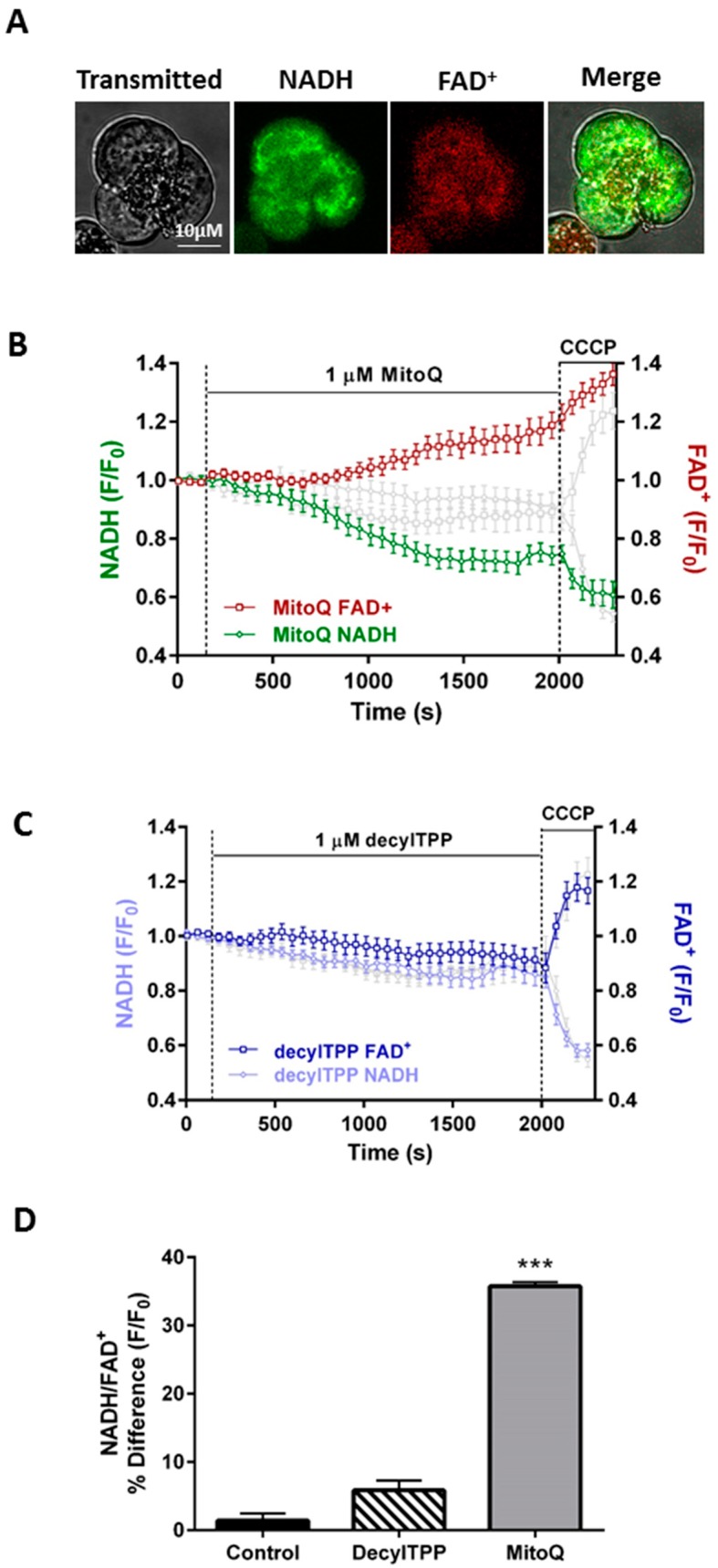
Effects of MitoQ and DecylTPP on the redox ratio. (**A**) Typical images showing the transmitted light (left) and mitochondrial localisation of NADH and FAD^+^ autofluorescence in a triplet of pancreatic acinar cells, measured simultaneously using confocal microscopy. (**B**) Effects of MitoQ and (**C**) DecylTPP (1 µM) and CCCP (10 µM) on NADH and FAD^+^ levels, expressed as normalised values from control (F/F_0_). Time-matched control recordings are shown in grey. CCCP was applied to show a maximal effect. (**D**) Effects of MitoQ and DecylTPP (1 µM) on the redox ratio (NADH/FAD^+^) expressed as a percentage of basal values. Traces are averages of >19 cells from at least 3 animals. All data shown are mean ± SEM. The bar chart is presented as a percentage change from the baseline until the end of treatment. *** *p* < 0.001 NaHEPES control vs 1μM MitoQ.

**Figure 4 ijms-20-01700-f004:**
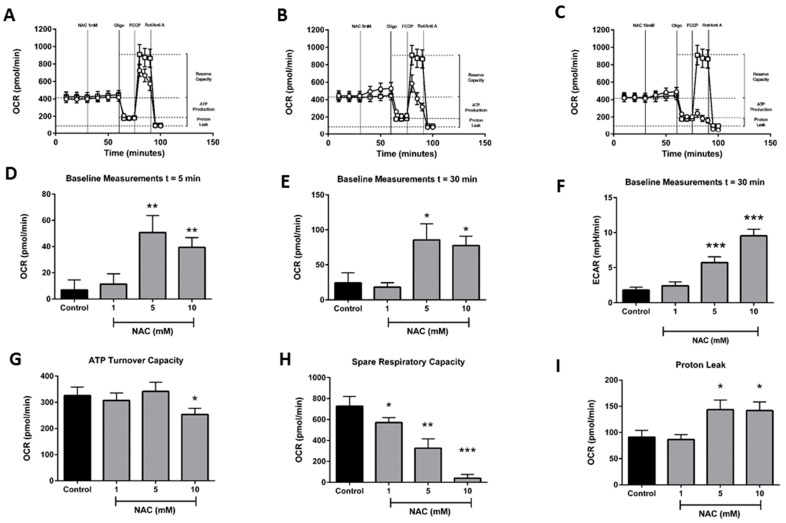
Effects of *N*-acetylcysteine on mitochondrial bioenergetics. Plots of respiratory control experiments showing the time-dependent effects of *N*-acetylcysteine (open circles) (**A**) 1 mM, (**B**) 5 mM, and (**C**) 10 mM on OCR changes compared to control (open squares). A respiratory function ‘stress’ test was carried out using sequential addition of oligomycin (Oligo, 1 μg/mL), FCCP (0.3 μM), and rotenone/antimycinA combined (Rot/AntiA, 1 μM). Bar charts showing the effects of *N*-acetylcysteine (1–10 mM) on OCR at (**D**) 5 min, (**E**) 30 min and ECAR at (**F**) 30 min, and (**G**) ATP turnover capacity, (**H**) proton leak and (**I**) spare respiratory capacity. Data shown as means ± SEM (*n* = 3). * *p* < 0.05, ** *p* < 0.01, *** *p* < 0.001.

**Figure 5 ijms-20-01700-f005:**
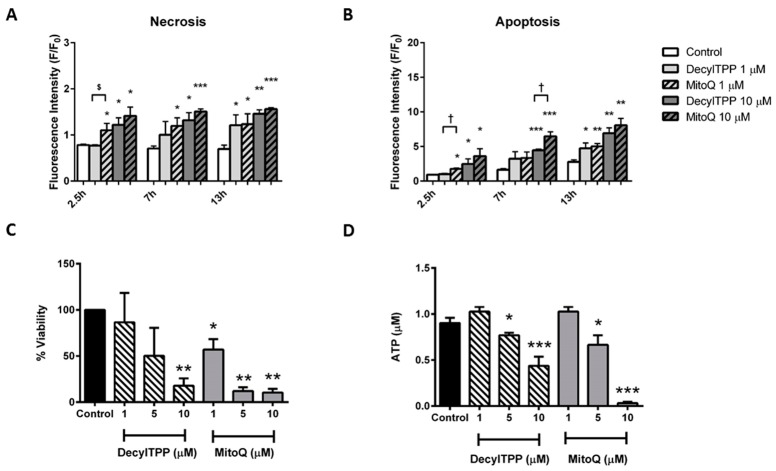
Effects of MitoQ and Decyl TPP on cell fate. Effects of MitoQ and DecylTPP (1–10 µM) on (**A**) necrosis measured by propidium iodide (**B**) apoptosis measured by caspase 3/7. Data have been normalised to the initial fluorescence reading at *t* = 0 and expressed as F/F_0_. All data shown are mean ± SEM (*n* = 3). (**C**) Cell viability measured via intracellular lactate dehydrogenase levels, with data shown as percent viability and (**D**) ATP levels (µM per 1 × 10^6^ cells) measured by a luciferase assay. All data shown as mean ± SEM (*n* = 3). * *p* < 0.05, ** *p* < 0.01, *** *p* < 0.001. ^†^
*p* < 0.05.

**Figure 6 ijms-20-01700-f006:**
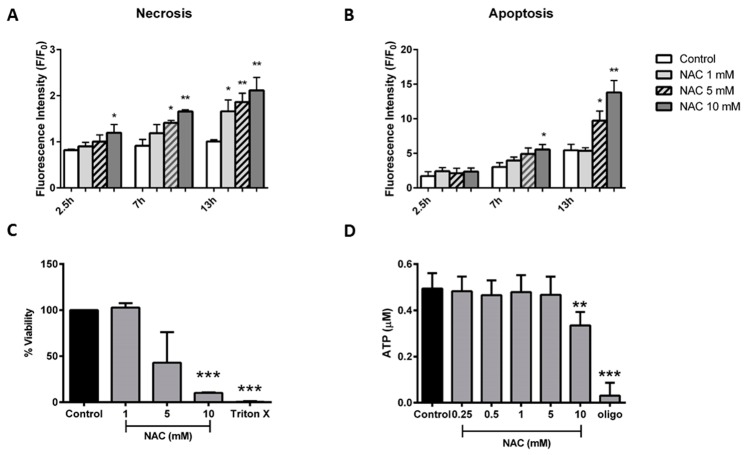
Effects of *N*-acetylcysteine on cell fate. Effects of *N*-acetylcysteine (NAC) (1–10 mM) on (**A**) necrosis measured by propidium iodide and (**B**) apoptosis measured by caspase 3/7. Data have been normalised to the initial fluorescence reading at *t* = 0 and expressed as F/F_0_. All data shown are mean ± SEM (*n* = 3). (**C**) Cell viability measured via intracellular lactate dehydrogenase levels; data shown as percent viability and (**D**) ATP levels (μM per 1 × 10^6^ cells) measured by luciferase assay. All data shown as mean ± SEM (*n* = 3). * *p* < 0.05, ** *p* < 0.01, *** *p* < 0.001.

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
