# Peer review of "Mitochondrial Targeting of Antioxidants Alters Pancreatic Acinar Cell Bioenergetics and Determines Cell Fate"

_ijms, 2019, doi:10.3390/ijms20071700_

Reviewer 1 Report

In this study, Armstrong et al. focused on how ROS levels affect the bioenergetic profile and cell fate of pancreatic acinar cells. The Authors showed that both general antioxidant (NAC) and mitochondrial-targeting antioxidant (MitoQ) alter the pancreatic acinar cell bioenergetics by increasing the basal oxidative respiration and reducing spare respiratory capacity. Additionally, the treatment of antioxidants triggers necrosis and apoptosis and therefore, negatively affects the cell viability. The experiments were rigorously performed, but I have a few concerns to be addressed:

1) Assays that directly determine the ROS levels in the cells should be included to show whether the effects of antioxidants are due to the alteration of ROS levels. Additionally, DecylTPP has been reported to be able to induce oxidative stress (Schibler et al, 2016), therefore, it is important to show that decylTPP has no impact on the ROS levels in the experiments before decylTPP could be used as a non-antioxidant control. 

2) Treatment of MitoQ elevated both baseline oxidative phosphorylation and glycolysis, how to explain the reduction of APT levels?

3) Figure 3B and 3C were not properly annotated. In the figure legend, both panels contain MitoQ and DecylTPP, while the figures were labeled with either mitoQ(1uM) or  DecylTPP(1uM) at the top of the each panel. And what are the grey plots in figure 3B? I assume those are NADH and FAD after DecylTPP treatment but they are not labeled. 

4) PI staining alone is not indicative of necrosis, late stage apoptotic cells can also be PI positive.

Author Response

Referee 1

In this study, Armstrong et al. focused on how ROS levels affect the bioenergetic profile and cell fate of pancreatic acinar cells. The Authors showed that both general antioxidant (NAC) and mitochondrial-targeting antioxidant (MitoQ) alter the pancreatic acinar cell bioenergetics by increasing the basal oxidative respiration and reducing spare respiratory capacity. Additionally, the treatment of antioxidants triggers necrosis and apoptosis and therefore, negatively affects the cell viability. The experiments were rigorously performed, but I have a few concerns to be addressed:

 The authors appreciate that the reviewer has noted the rigorous approach adopted in this evaluation and their positive assessment of the study.

 1) Assays that directly determine the ROS levels in the cells should be included to show whether the effects of antioxidants are due to the alteration of ROS levels. Additionally, DecylTPP has been reported to be able to induce oxidative stress (Schibler et al, 2016), therefore, it is important to show that decylTPP has no impact on the ROS levels in the experiments before decylTPP could be used as a non-antioxidant control.

 We agree that this is an important aspect to clarify. The authors have previously published results demonstrating that DecylTPP (and MitoQ) did not increase ROS levels per se in mouse pancreatic acinar cells at 1µM [1]. We have now performed additional experiments which demonstrate that there was no significant elevation of ROS induced by 10µM MitoQ or DecylTPP in this cell type (n=3). Thus our data confirm that DecylTPP does not cause its effects via a generation of oxidative stress in pancreatic acinar cells and this information has now been included as a Supplementary Figure 1.

 2) Treatment of MitoQ elevated both baseline oxidative phosphorylation and glycolysis, how to explain the reduction of APT levels?

 The ATP turnover capacity (observed after oligomycin application) was greatly reduced by MitoQ (and by DecylTPP); a fall in total cellular ATP (detected via luciferase) is consistent with this effect and further reflected by a proportion of PACs undergoing cell death. A compensatory boost of ECAR occurred in response to both MitoQ and DecylTPP, indicating an attempt to compensate by increasing glycolytic ATP production, although ultimately inhibition of Oxphos-mediated ATP production was paramount and cellular ATP levels fell with time. The elevated baseline respiration in response to MitoQ (but not Decyl TPP) may reflect relief from a tonic inhibition of basal OCR by endogenous ROS; the inhibition of ATP turnover capacity (shared by both MitoQ and DecylTPP) would appear to be mechanistically distinct.

 3) Figure 3B and 3C were not properly annotated. In the figure legend, both panels contain MitoQ and DecylTPP, while the figures were labeled with either mitoQ(1uM) or  DecylTPP(1uM) at the top of the each panel. And what are the grey plots in figure 3B? I assume those are NADH and FAD after DecylTPP treatment but they are not labeled. 

 We apologise for the incorrect legend. This has now been amended to reflect the data presented in Figure 3.

 4) PI staining alone is not indicative of necrosis, late stage apoptotic cells can also be PI positive.

 The authors agree that late apoptotic cells may also be detected by PI staining. However, PI staining of necrotic cells is an established method for evaluation of necrosis; a differentiation between late apoptotic and necrotic cell death, the basis of which is still incompletely understood, was not a primary objective of this study.

 References

 1.            Huang, W.; Cash, N.; Wen, L.; Szatmary, P.; Mukherjee, R.; Armstrong, J.; Chvanov, M.; Tepikin, A. V.; Murphy, M. P.; Sutton, R.; Criddle, D. N., Effects of the mitochondria-targeted antioxidant mitoquinone in murine acute pancreatitis. Mediators Inflamm 2015, 2015, 901780.

Reviewer 2 Report

 The manuscript “Mitochondrial targeting of antioxidants alters pancreatic acinar cell bioenergetics and determines cell fate” by Armstrong at al addresses the mitochondrial effects of antioxidants that are supposed to prevent oxidative stress resulting from mitochondria injury. The problem scoped out in this manuscript is important, as for years the protective abilities of antioxidants were overestimated, while experimental data and clinical trials demonstrated the limited capacity of exogenous antioxidants against elevated ROS.  Although comparative data were obtained regarding the effects of MitoQ, DecylTPP and the classical compound NAC on mitochondria OxPhos, the chosen research strategy seems to have misled authors to the conclusions which are hard to believe.

The rationale behind the selected doses 1, 5 and 10 μM for MitoQ and DecylTPP is unclear. The viability data shows up somewhere in the middle of the story, in Figure 5C, but not in the beginning of the results. This data clearly demonstrates, especially for MitQ, that 5 and 10 μM are subtoxic concentrations, causing roughly 80-85% death. Therefore, such high concentrations that cause high percentage of cell death are unacceptable to address the mechanisms. They reflect the agent’s toxicity. The fact that both death types, necrosis and apoptosis, develop over time (Figure 5A,B) and the level of ATP for both MitoQ and DecylTPP are surprisingly not dramatically low at 5 μM (Figure 5D), while this concentration causes almost complete death of cell population (Figure 5C) tells that the complex of disruptive processes occurs in the cells at 5 and 10 μM doses, causing acute toxicity. This fact makes the rest of the data pointless.

However, I would like to highlight some other observations.

-        For those readers who might see the studied compounds MitoQ and DecylTPP for the first time, it would be useful to get some information on their chemical structure, the common  and distinct features.

-        The major focus of the study is the effect of agents on mitochondria OxPhos. The use of classical terminology, such as coupling/uncoupling, inhibition of ATP synthesis, maximal respiratory capacity etc would be more appropriate rather than reliance on the manufacturer (Seahorse) provided protocols. What exactly does the spare respiratory capacity mean and how could the effects of the agents be interpreted in the light of this capacity? (line 85)

-        If the oligomycin is applied, that means that the ATP synthase is blocked. What kind of non-specific targeting do the authors mean (lines 82-83)?

-        Increase in basal respiration is not necessary but still could refer to uncoupling of respiration (line 74). Yet, the two sets of data, respiration and extracellular acidification, remain on their own with no further explanation or attempts to link them (lines 74-80).

-        Unfortunately, rotenon and antimycin were added together, I assume again it was done simply to follow the manufacturer protocol. This does not allow to distinguish between the possible damaging effects of the agents on complex I, II or III.

-        The data in Figure 2 are also confusing:

Thus, in graph B DecylTPP does not induce significant stimulation of respiration in contrast to control. However, in graph F both concentrations 5 and 10 μM promote strong proton leak of the same level as MitoQ. Why such strong uncoupling does not result in stimulation of respiration? Graph G does not provide much information, it is rather confusing and could be omitted.

-        The data on NADH/FAD are shown with limited connection to the respirometric results and potential mitochondrial processes that could be involved.

 The Discussion contains lots of references on previously published works. However, since many effects are cell type dependent, the key factors are better to be addressed in the current work. For example, the key mitochondria parameter, the membrane potential, could be altered differently than it was observed in breast and lung cancer cells (line 222). And this parameter is important to address the effects of MitoQ, DecylTPP, and NAC on OxPhos in cells studied in the current work. Similarly, as the studied agents are considered antioxidants, it would be helpful to demonstrate whether they attenuate ROS or rather, maybe they bust oxidative stress, especially at the chosen concentrations causing acute toxicity. Therefore, the ROS related discussion is unsupported.

The authors assumption regarding the non-specific targeting of charged acyl moieties of mitochondria (lines 233-246) is also unsupported. 

There are few concerns regarding methods:

-        For detection of apoptosis and necrosis authors seeded cells at the density 300,000 cells per well. What is the size of acinar cells? It is hard to believe that on the square area of 96-well plate will fit 300,000 cells. Do they mean 30,000? Even the latter is huge. It is known that the space limit affects cell mitochondria membrane potential and induces death signals if cells overgrown.

-        For intracellular ATP detection authors used buffer A that mimics intracellular environment. Why did they use non-physiological, more basic pH 7.8, as it is well known that the cytosolic pH is close to neutral, about 7.1.

Unfortunately, although the authors did attempt to address an important topic, the power of exogenous antioxidants to attenuate oxidative stress, which is currently under dispute, as more and more data show rather their limited ability, the chosen experimental approach did not allow them to cover the task. Overall, the assumptions and conclusions are loose and hardly supported by the own data.

Author Response

Referee 2

 The manuscript “Mitochondrial targeting of antioxidants alters pancreatic acinar cell bioenergetics and determines cell fate” by Armstrong at al addresses the mitochondrial effects of antioxidants that are supposed to prevent oxidative stress resulting from mitochondria injury. The problem scoped out in this manuscript is important, as for years the protective abilities of antioxidants were overestimated, while experimental data and clinical trials demonstrated the limited capacity of exogenous antioxidants against elevated ROS.  Although comparative data were obtained regarding the effects of MitoQ, DecylTPP and the classical compound NAC on mitochondria OxPhos, the chosen research strategy seems to have misled authors to the conclusions which are hard to believe.

Unfortunately, although the authors did attempt to address an important topic, the power of exogenous antioxidants to attenuate oxidative stress, which is currently under dispute, as more and more data show rather their limited ability, the chosen experimental approach did not allow them to cover the task. Overall, the assumptions and conclusions are loose and hardly supported by the own data.

The authors are pleased that the reviewer noted the importance of the topic addressed in the present study, however, disagree that the strategy adopted has led to erroneous conclusions.

The rationale behind the selected doses 1, 5 and 10 μM for MitoQ and DecylTPP is unclear. The viability data shows up somewhere in the middle of the story, in Figure 5C, but not in the beginning of the results. This data clearly demonstrates, especially for MitQ, that 5 and 10 μM are subtoxic concentrations, causing roughly 80-85% death. Therefore, such high concentrations that cause high percentage of cell death are unacceptable to address the mechanisms. They reflect the agent’s toxicity. The fact that both death types, necrosis and apoptosis, develop over time (Figure 5A,B) and the level of ATP for both MitoQ and DecylTPP are surprisingly not dramatically low at 5 μM (Figure 5D), while this concentration causes almost complete death of cell population (Figure 5C) tells that the complex of disruptive processes occurs in the cells at 5 and 10 μM doses, causing acute toxicity. This fact makes the rest of the data pointless.

The concentrations used were chosen with respect to our previous study in which a lack of protection by MitoQ was apparent in acute pancreatitis models, although some mixed effects were detected [1]. In addition, a previous publication used µM MitoQ to evaluate the effects of antioxidants on calcium signalling in this cell type [2], whilst other studies have applied 2 µM MitoQ to MDA-MB-231 and H23 cells [3] and 10 µM MitoQ to Hela cells [4]. The current investigation aimed to elucidate effects of MitoQ on pancreatic acinar mitochondrial bioenergetics in order to cast light on prior results. However, rather than the detrimental actions detected in the present study being unimportant, these new data clearly demonstrate significant alterations of mitochondrial bioenergetics that were directly related to mitochondrial targeting per se and to antioxidant activity within mitochondria. These actions resulted in concentration-dependent ATP depletion and cell death. Understanding mechanisms of cellular toxicity is vitally important, especially when such drugs are subject to clinical evaluation as potential therapies. Although the reviewer states that changes to ATP were not dramatically low at ≥ 5 µM, they were significantly different from controls. The ATP (luciferase-based) measurements reflect changes in a population of cells, as do the plate-reader evaluations of cell death; a precise correlation between ATP fall and necrosis in individual cells would not be obtained using this methodology. Nevertheless, induction of pancreatic acinar cell death by MitoQ and DecylTPP was coincident with alterations of bioenergetics resulting in a significant cellular ATP decrease.

 However, I would like to highlight some other observations.

-        For those readers who might see the studied compounds MitoQ and DecylTPP for the first time, it would be useful to get some information on their chemical structure, the common  and distinct features.

 Thank you for this helpful suggestion. We have now pointed interested readers to previously published structural information on chemical composition [5] (Introduction Line 62).

 -        The major focus of the study is the effect of agents on mitochondria OxPhos. The use of classical terminology, such as coupling/uncoupling, inhibition of ATP synthesis, maximal respiratory capacity etc would be more appropriate rather than reliance on the manufacturer (Seahorse) provided protocols. What exactly does the spare respiratory capacity mean and how could the effects of the agents be interpreted in the light of this capacity? (line 85)

 Seahorse flux analysis, although a relatively recent innovation, is a widely published method considered by experts in the field as a prime approach for assessing mitochondrial dysfunction in cells [6]. The terminology used reflects the protocols developed; thus spare respiratory capacity is a term applied to indicate the difference between basal respiration (OCR) and maximal respiration after FCCP application. Importantly, increasing evidence indicates that this parameter is diminished in disease states (please refer to the Discussion); MitoQ (and NAC) exerted a strong reduction of SRC which may have relevance to its lack of efficacy to protect in acute pancreatitis in vivo models, although the underlying mechanism is currently unknown.

 -        If the oligomycin is applied, that means that the ATP synthase is blocked. What kind of non-specific targeting do the authors mean (lines 82-83)?

 A possible basis for non-specific targeting effects (i.e. common to both MitoQ and DecylTPP) was addressed in the Discussion; “…it was proposed that the DecylTPP moiety undergoes rapid cycling between the mitochondrial matrix and cytosol thereby uncoupling mitochondria, consistent with inhibition of ATP turnover capacity. The effect we observed in pancreatic acinar cells was accompanied by a marked elevation of proton leak with both MitoQ and DecylTPP, suggesting a compromise of mitochondrial membrane integrity by insertion of the DecylTPP moiety. DecylTPP has recently been shown to directly increase the permeability of artificial liposomes; in that study both MitoQ and DecylTPP caused mitochondrial membrane depolarisation and swelling of kidney tissue that was dependent on the acyl chain.”

 -        Increase in basal respiration is not necessary but still could refer to uncoupling of respiration (line 74). Yet, the two sets of data, respiration and extracellular acidification, remain on their own with no further explanation or attempts to link them (lines 74-80).

 A link between respiration and extracellular acidification was dealt with in the Discussion section; “…a robust increase of pancreatic acinar cell ECAR in response to MitoQ, DecylTPP and NAC occurred concomitantly with inhibition of ATP turnover, suggesting a compensatory shift in metabolism towards glycolysis when oxidative phosphorylation was inhibited”. We have now provided an additional comment in the Discussion citing a recent study in which such a glycolytic shift in response to MitoQ was detected: “An impairment of oxidative phosphorylation coincident with an increased ECAR in response to MitoQ has recently been shown in MDA-MB-231 cells suggesting a switch in energy production to glycolysis in this cell line [29].”

 -        Unfortunately, rotenon and antimycin were added together, I assume again it was done simply to follow the manufacturer protocol. This does not allow to distinguish between the possible damaging effects of the agents on complex I, II or III.

 The addition of rotenone and antimycin together is a standard component of the mitochondrial respiratory function (“stress”) test, used in many published investigations, including those of MitoQ [5]. Interactions of MitoQ and analogues with individual respiratory chain complexes have previously been explored [7]. Although a detailed evaluation of MitoQ on ETC components in pancreatic acinar cells might be interesting and the basis for a future investigation, we agree with the reviewer that possible damaging effects of the agents on individual complexes was not possible to determine within the current protocols.

 -        The data in Figure 2 are also confusing:

Thus, in graph B DecylTPP does not induce significant stimulation of respiration in contrast to control. However, in graph F both concentrations 5 and 10 μM promote strong proton leak of the same level as MitoQ. Why such strong uncoupling does not result in stimulation of respiration? Graph G does not provide much information, it is rather confusing and could be omitted.

 The difference between MitoQ and DecylTPP on basal OCR likely reflects a specific antioxidant effect to inhibit a tonic suppression of basal respiration by endogenous ROS (this effect of MitoQ was shared by NAC). Both MitoQ and DecylTPP, however, greatly reduced the OCR response to oligomycin indicating a reduction of ATP turnover (mirrored by a decrease of total cellular ATP). The elevated proton leak, calculated as the difference between the oligomycin plateau response and that caused by subsequent rotenone/antimycin addition, may suggest an effect on OCR that is somehow exacerbated under conditions of ATP synthase inhibition, although the mechanism underlying these findings is currently unclear.

 Graph G is necessary since it shows the significant decrease of Spare Respiratory Capacity induced by MitoQ, an action shared by NAC but not by DecylTPP.

 -        The data on NADH/FAD are shown with limited connection to the respirometric results and potential mitochondrial processes that could be involved.

 Confocal microscopy was employed to provide a complementary, but distinct, methodology with which to examine bioenergetics changes in acinar cells. The results here show that MitoQ induced a profound decrease of NADH, with mirrored increase of FAD+, that is consistent with the inhibitory effects observed in the Seahorse flux analysis of bioenergetics.

 The Discussion contains lots of references on previously published works. However, since many effects are cell type dependent, the key factors are better to be addressed in the current work. For example, the key mitochondria parameter, the membrane potential, could be altered differently than it was observed in breast and lung cancer cells (line 222). And this parameter is important to address the effects of MitoQ, DecylTPP, and NAC on OxPhos in cells studied in the current work. Similarly, as the studied agents are considered antioxidants, it would be helpful to demonstrate whether they attenuate ROS or rather, maybe they bust oxidative stress, especially at the chosen concentrations causing acute toxicity. Therefore, the ROS related discussion is unsupported.

The authors assumption regarding the non-specific targeting of charged acyl moieties of mitochondria (lines 233-246) is also unsupported. 

 The novel results of our study in pancreatic acinar cells need to be addressed within the context of established literature, comprised of investigations of MitoQ in diverse cell types. We agree that mitochondrial membrane potential is an important parameter to investigate; this was already done (and cited in the manuscript) in pancreatic acinar cells as part of our previous publication which showed a decrease of Ψ∆m in response to MitoQ [1]. This result is consistent with the bioenergetics changes reported in the present study and with effects shown in the publications cited using breast and lung cancer cells. Furthermore, our prior study showed that MitoQ, but not decylTPP, inhibited the rise of ROS induced by application of H2O2 [1] consistent with the reported actions of MitoQ in other cell types.

 Both MitoQ and DecylTPP caused large reductions of ATP turnover at the same concentrations; this clearly indicates non-specific effects of mitochondrial targeting that involves the targeting moiety i.e. the charged acyl chain. Published evidence that might explain this action was included in the Discussion.

 There are few concerns regarding methods:

-        For detection of apoptosis and necrosis authors seeded cells at the density 300,000 cells per well. What is the size of acinar cells? It is hard to believe that on the square area of 96-well plate will fit 300,000 cells. Do they mean 30,000? Even the latter is huge. It is known that the space limit affects cell mitochondria membrane potential and induces death signals if cells overgrown.

 It is important to note that these are primary (i.e. freshly isolated) cells from mouse pancreas and not a cultured cell line. The cells are not dividing and therefore not overgrown under conditions of the assay. Our previous experience with cell death assays [8] indicates that the PACs fit comfortably onto the plate wells, producing reliable and reproducible measurements of apoptosis and necrosis.

 -        For intracellular ATP detection authors used buffer A that mimics intracellular environment. Why did they use non-physiological, more basic pH 7.8, as it is well known that the cytosolic pH is close to neutral, about 7.1. 

This is a standard, widely-used and published methodology [9]; the study has followed the manufacturer’s instructions appropriately.

References

 1.            Huang, W.; Cash, N.; Wen, L.; Szatmary, P.; Mukherjee, R.; Armstrong, J.; Chvanov, M.; Tepikin, A. V.; Murphy, M. P.; Sutton, R.; Criddle, D. N., Effects of the mitochondria-targeted antioxidant mitoquinone in murine acute pancreatitis. Mediators Inflamm 2015, 2015, 901780.

2.            Camello-Almaraz, M. C.; Pozo, M. J.; Murphy, M. P.; Camello, P. J., Mitochondrial production of oxidants is necessary for physiological calcium oscillations. J Cell Physiol 2006, 206, (2), 487-94.

3.            Pokrzywinski, K. L.; Biel, T. G.; Kryndushkin, D.; Rao, V. A., Therapeutic Targeting of the Mitochondria Initiates Excessive Superoxide Production and Mitochondrial Depolarization Causing Decreased mtDNA Integrity. PLoS One 2016, 11, (12), e0168283.

4.            Leo, S.; Szabadkai, G.; Rizzuto, R., The mitochondrial antioxidants MitoE(2) and MitoQ(10) increase mitochondrial Ca(2+) load upon cell stimulation by inhibiting Ca(2+) efflux from the organelle. Ann N Y Acad Sci 2008, 1147, 264-74.

5.            Reily, C.; Mitchell, T.; Chacko, B. K.; Benavides, G.; Murphy, M. P.; Darley-Usmar, V., Mitochondrially targeted compounds and their impact on cellular bioenergetics. Redox Biol 2013, 1, (1), 86-93.

6.            Brand, M. D.; Nicholls, D. G., Assessing mitochondrial dysfunction in cells. Biochem J 2011, 435, (2), 297-312.

7.            James, A. M.; Cocheme, H. M.; Smith, R. A.; Murphy, M. P., Interactions of mitochondria-targeted and untargeted ubiquinones with the mitochondrial respiratory chain and reactive oxygen species. Implications for the use of exogenous ubiquinones as therapies and experimental tools. J Biol Chem 2005, 280, (22), 21295-312.

8.            Armstrong, J. A.; Cash, N. J.; Ouyang, Y.; Morton, J. C.; Chvanov, M.; Latawiec, D.; Awais, M.; Tepikin, A. V.; Sutton, R.; Criddle, D. N., Oxidative stress alters mitochondrial bioenergetics and modifies pancreatic cell death independently of cyclophilin D, resulting in an apoptosis-to-necrosis shift. J Biol Chem 2018.

9.            Kiesslich, T.; Benno Oberdanner, C.; Krammer, B.; Plaetzer, K., Fast and reliable determination of intracellular ATP from cells cultured in 96-well microplates. J Biochem Biophys Methods 2003, 57, (3), 247-51.

Reviewer 3 Report

The authors used relatively high concentration of MitoQ (1, 5, 10 µM) and NAC (1, 5, 10 mM). Aldini et al. published that clinically relevant human plasma concentrations of NAC are 0–1000 µg/mL (Aldini et al., Free Radic Res. 52:751-762, 2018), so the authors used 1, 5, 10 mM of NAC fits in the range of its clinical plasma concentration. However, regarding MitoQ concentration, Smith and Murphy described that in human Phase 1 trials MitoQ showed good pharmacokinetic behavior with oral dosing at 80 mg (1 mg/kg) resulted in a plasma maxi1mal concentration of 33.15 ng/mL after ∼1 h (Smith and Murphy, Ann N Y Acad Sci. 1201:96-103. 2010). And, Solesio et al. used 50 nM of MitoQ in their paper (Solesio et al., Biochim Biophys Acta. 2013 Jan;1832(1):174-82.). The authors used relatively high concentration of MitoQ in this manuscript. The authors should discuss the differences of the concentration regarding MitoQ.

1 µM = 678.82 µg/L

= 678.82 ng/mL

5 µM = 33941 ng/mL

= 33.941 µg/mL

10 µM = 67,89 µg/mL

 P8, line 2 from bottom: “recent evidence demonstrates that ROS participate in physiological signalling events [34, 35].” Please cite the following article,

Indo, H. P.; Hawkins, C. L.; Nakanishi, I.; Matsumoto, K.; Matsui, H.; Suenaga, S.; Davies, M. J.: St Clair, D. K.; Ozawa, T.; Majima, H. J. Role of Mitochondrial Reactive Oxygen Species in the Activation of Cellular Signals, Molecules and Function. In Pharmacology of Mitochondria; Handb Exp Pharmacol (HEP), Eds.; Singh, H., Sheu S.-S.; Springer Nature Switzerland AG: Basel, 2017; Volume 240, pp. 439-456. doi: 10.1007/164_2016_117.

 Author Response

Referee 3

The authors used relatively high concentration of MitoQ (1, 5, 10 µM) and NAC (1, 5, 10 mM). Aldini et al. published that clinically relevant human plasma concentrations of NAC are 0–1000 µg/mL (Aldini et al., Free Radic Res. 52:751-762, 2018), so the authors used 1, 5, 10 mM of NAC fits in the range of its clinical plasma concentration. However, regarding MitoQ concentration, Smith and Murphy described that in human Phase 1 trials MitoQ showed good pharmacokinetic behavior with oral dosing at 80 mg (1 mg/kg) resulted in a plasma maximal concentration of 33.15 ng/mL after 1 h (Smith and Murphy, Ann N Y Acad Sci. 1201:96-103. 2010). And, Solesio et al. used 50 nM of MitoQ in their paper (Solesio et al., Biochim Biophys Acta. 2013 Jan;1832(1):174-82.). The authors used relatively high concentration of MitoQ in this manuscript. The authors should discuss the differences of the concentration regarding MitoQ.

1 µM = 678.82 µg/L = 678.82 ng/mL , 5 µM = 33941 ng/mL = 33.941 µg/mL, 10 µM = 67,89 µg/mL

Thank you for this valuable comment. The authors appreciate that the concentrations of MitoQ used in the study were relatively high, although a previous publication in this cell type used micromolar MitoQ to evaluate the effects of antioxidants on calcium signalling [1], whilst other studies have applied 2 µM MitoQ to MDA-MB-231 and H23 cells [2] and 10 µM MitoQ to Hela cells [3].The concentration range in the present study was chosen to be consistent with our previous investigation in which MitoQ treatment was largely ineffective in models of acute pancreatitis [4] as a basis to further elucidate its cellular actions. The authors acknowledge the importance of including discussion of the concentrations used, since they bear relevance to on-going therapeutic evaluations of MitoQ, and have amended the Discussion accordingly: “It should be noted that the concentrations of MitoQ investigated in the present study may be higher than those achieved during therapeutic application. Previous studies have suggested that the intra-mitochondrial concentration of MitoQ was approximately 0.7 µM in mice fed 500 µM MitoQ in drinking water for 4-6 months [27], while intravenous injection of 750 nM MitoQ was well-tolerated, although toxic effects were evident at 1 µM [50].”

P8, line 2 from bottom: “recent evidence demonstrates that ROS participate in physiological signalling events [34, 35].” Please cite the following article,

Indo, H. P.; Hawkins, C. L.; Nakanishi, I.; Matsumoto, K.; Matsui, H.; Suenaga, S.; Davies, M. J.: St Clair, D. K.; Ozawa, T.; Majima, H. J. Role of Mitochondrial Reactive Oxygen Species in the Activation of Cellular Signals, Molecules and Function. In Pharmacology of Mitochondria; Handb Exp Pharmacol (HEP), Eds.; Singh, H., Sheu S.-S.; Springer Nature Switzerland AG: Basel, 2017; Volume 240, pp. 439-456. doi: 10.1007/164_2016_117.

The authors are grateful for this indication and have now cited the article.

 References

 1.            Camello-Almaraz, M. C.; Pozo, M. J.; Murphy, M. P.; Camello, P. J., Mitochondrial production of oxidants is necessary for physiological calcium oscillations. J Cell Physiol 2006, 206, (2), 487-94.

2.            Pokrzywinski, K. L.; Biel, T. G.; Kryndushkin, D.; Rao, V. A., Therapeutic Targeting of the Mitochondria Initiates Excessive Superoxide Production and Mitochondrial Depolarization Causing Decreased mtDNA Integrity. PLoS One 2016, 11, (12), e0168283.

3.            Leo, S.; Szabadkai, G.; Rizzuto, R., The mitochondrial antioxidants MitoE(2) and MitoQ(10) increase mitochondrial Ca(2+) load upon cell stimulation by inhibiting Ca(2+) efflux from the organelle. Ann N Y Acad Sci 2008, 1147, 264-74.

4.            Huang, W.; Cash, N.; Wen, L.; Szatmary, P.; Mukherjee, R.; Armstrong, J.; Chvanov, M.; Tepikin, A. V.; Murphy, M. P.; Sutton, R.; Criddle, D. N., Effects of the mitochondria-targeted antioxidant mitoquinone in murine acute pancreatitis. Mediators Inflamm 2015, 2015, 901780. 

Round  2

Reviewer 1 Report

My questions have been very well addressed. 

Author Response

We wish to thank the reviewer for their constructive criticism of the study and positive response to our revision.

Reviewer 3 Report

Tha manuscript became much better. The manuscript is acceptable.

Author Response

We wish to thank the reviewer for their constructive criticism to improve the study and positive response to our revision.